# Small Spacecraft Payload Study for X-ray Astrophysics including GRB Science

Vladimír Dániel [1,*], Veronika Maršíková [2], Rene Hudec [3,4], Ladislav Pína [2,5], Adolf Inneman [2] and Karel Pelc [1]

1. Czech Aerospace Research Centre, 199 05 Prague, Czech Republic; pelc@vzlu.cz
2. Rigaku Innovative Technologies Europe, 142 00 Prague, Czech Republic; veronika.marsikova@rigaku.com (V.M.); ladislav.pina@rigaku.com (L.P.); adolf.inneman@rigaku.com (A.I.)
3. Faculty of Electrical Engineering, Czech Technical University in Prague, 166 27 Prague, Czech Republic; rene.hudec@gmail.com
4. Engelhardt Observatory, Kazan Federal University, 420008 Kazan, Russia
5. Faculty of Nuclear Science, Czech Technical University in Prague, 115 19 Prague, Czech Republic
* Correspondence: daniel@vzlu.cz; Tel.: +420-725-380-564

**Abstract:** This paper presents microsatellite spacecraft payload study for prompt observation of transient astrophysical objects in X-ray energy range. By combining telescope concepts and miniaturized detectors, the small spacecraft will be able to probe the X-ray temporal emissions of bright events such as Gamma-Ray Bursts (GRBs), X-ray transients or the electromagnetic counterparts of Gravitational Wave Events (GWEs), but also short and long term observations of other types of variable X-ray sources. The spacecraft is based on the CubeSat nanosatellite platform with a volume of 16U. The spacecraft carries two types of X-ray telescopes onboard. The first is intended for X-ray transient monitoring and localization, and the second for detailed spectroscopic observation. The X-ray monitor/localization telescope with wide field of view of several arc degrees is used for localization and flux measurement of X-ray transients, as well as for permanent monitoring of Galactic center area. This telescope is based on Lobster Eye X-ray optics together with pixel detector based on the Timepix3 Quad detector. Rapid follow-up observation by soft X-ray spectroscopy is enabled by a second X-ray spectroscopic telescope with limited FOV (Field of View) of several arcmins with no spatial and/or angular resolution. The spectroscopic telescope uses condenser optics based on replicated parabolic total reflection system (or, alternatively, Wolter system) and a Ketek X-ray SDD detector with energy resolution of about 130 eV as a detector. In addition to technical and instrumental aspects, observational strategy and astrophysical issues and justifications are also addressed in the paper.

**Keywords:** X-rays; X-ray astronomy; X-ray transients; GRBs; CubeSat; X-ray optics

## 1. Introduction

A large number of high-energy transient events can be observed in the sky. One type of these transients are Gamma-Ray Bursts (GRBs), discovered in 1967 by the Vela military satellites [1]. They are among the most extreme explosive events ever observed, momentarily outshining any other object in sky. Examples of other transients and variable sources observable in X-rays are novae and supernovae, transient binary X-ray sources, cataclysmic variables, soft X-ray transients, and several types of extra-galactic sources. New astronomical objects of unknown types are also being observed and classified as Exotic X-ray transients. In spite of large efforts and numerous observations, many open questions about the details of their physics remain.

Despite past and recent progress in X-ray astrophysics, monitoring missions are still limited, especially those operated on soft X-rays. On the other hand, the X-ray sky is very rich in highly variable, transient, and flaring sources, so there is an obvious need for more dedicated satellites and more extended sky monitoring.

Due to extended competitions with other projects and proposals, e.g., in the European Space Agency (ESA), it is difficult to have new X-ray astronomy missions approved (good examples are LOFT and THESEUS, which were preselected but finally not accepted for realization). This gap can be filled to some extent by small missions based on CubeSat technologies, enabled by the recent rapid progress in CubeSat-related technologies.

Currently, more than 10 space missions to discover, localize, and observe celestial X-ray sources are in operation. We can divide the instruments to monitors ("prospectors") and "observers" (instrumentation for detailed study). Various scientific instruments for X-ray transient monitoring are used with various source localization ability. The follow-up observation of X-ray transients is performed using space (as well as ground based) telescopes as "observers". To eliminate the transition and processing time between prospector (monitor) and observer (spectroscopic telescope) payload, they are sometimes both on the same spacecraft. In some special cases, such as the observation of GWE (Gravitational Wave Experiment) on Earth, only the "observer" mission obtains new importance. For GRBs, the rapid (in this case, minutes or less) follow-up observation of the GRBs represents the crucial mission feature to describe the initial part of the afterglow [2,3]. Recent large spacecrafts of high mass are usually not able to perform rapid orientation changes (up to dozens of degrees per second) to point the telescope to the required orientation. The small satellites with much less mass can achieve this repointing much easier.

### 1.1. X-ray Survey Missions

Eight selected currently active spacecrafts/instruments that survey the sky for X-ray and high-energy transients are listed in Table 1. The most accurate positions, of the order of a few arcmin, are provided by INTEGRAL IBIS (Imager on Board the INTEGRAL Satellite) [4] and by Swift BAT (Burst Alert Telescope) [5]. These instruments have an FOV (Field of View) of several degrees and provide precise localization resolution of about tens of arcseconds. There are also survey missions working in higher energies such as AstroSat CZTI [6] with a Cadmium Zinc Telluride Imager, which enables detection of hard X-rays (60–380 keV) with limiting FOV of 4.5° and high angular resolution.

Wider FOV is provided from instruments such as the SuperAGILE instrument onboard AGILE mission [7], which has large FOV of about 2.5 sr and angular resolution of about 2° [8]. Furthermore, collimating systems can be used, as shown on the MAXI (Monitor of All-sky X-ray Image) instrument [9] implemented onboard ISS. The instrument is equipped with a composite structure of slit holes and slat collimators without a mirror system, and achieves flat FOV of 160° × 1.5° with comparable angular resolution of 1.5°. Another method is to use vector sensitive detectors, such as flat or line detectors and ideally use three or six of them to achieve full sky coverage. This system is used by the Fermi Gamma-ray Burst Monitor (GBM) [10], which can see about 60% of the sky; however, it can only localize transients with an accuracy to the order of five degrees. Similarly, the Hard X-ray Modulation Telescope (HXMT) onboard ISIGHT (The Interior Exploration using Seismic Investigations, Geodesy and Heat Transport) mission [11] has a comparable resolution of several degrees. The recent GECAM (Gravitational Wave Electromagnetic Counterpart All-sky Monitor) [12] Chinese space telescope to monitor the Gamma-Ray Bursts (GRBs) that coincide with gravitational wave events uses flat LaBr3Ce detectors. Detectors onboard GECAM are placed on the half-sphere surface. By having two satellites on opposite sides of Earth, all sky coverage is achieved.

Only two missions listed in Table 1 represent missions with primary goal to detect GRBs; namely, SWIFT BAT and Fermi GBM.

Lastly, there were also missions providing GRB measurements with omnidirectional sensitivity such as Wind–Konus [13]. For localization of GRBs, their observations must be combined with observations of spacecrafts located elsewhere. Nevertheless, this experiment represents a valuable contributor to the GCN (The Gamma-ray Coordinates Network) Network [14].

**Table 1.** Summary of currently active X-ray survey satellite instruments and missions. It includes all mission types: not only followers, but also detectors.

| Mission<br>Launch | Optics<br>Detectors | Energy<br>FoV<br>Ang. Res. FWHM |
|---|---|---|
| INTEGRAL IBIS<br>2002 | Coded Mask<br>CdTe | 15 keV–1000 keV<br>19°<br>12′ |
| INTEGRAL SPI<br>2002 | Coded Mask<br>Ge | 20 keV–10 MeV<br>16°<br>2.5° |
| SWIFT BAT<br>2004 | Mirror Coded Mask<br>CdTe | 15 keV–150 keV<br>1.4 sr<br>17′ |
| AGILE<br>2007 | Coded Mask<br>Silicon microstrip | 15 keV–50 eV<br>2.5 sr<br>2° |
| MAXI (on ISS)<br>2008 | Collimator (ASM)<br>Gas PC Solid St. C | 2 keV–30 keV<br>160° × 1.5°<br>1.5° |
| FERMI GBM<br>2008 | No<br>NaI | 8 keV–40 MeV<br>9.5 sr<br>5° |
| AstroSat CZTI<br>2015 | Coded Mask<br>CZT | 60 keV–380 keV<br>4.5°<br>8′ |
| GECAM GRD<br>2020 | No<br>CLaBr3Ce | 6 keV–5 MeV<br>All-sky<br>1° |

Majority of these missions are operating beyond their nominal lifetimes. New missions are needed to continue monitoring the sky and can also be used to detect and follow up the new types of targets such as the electromagnetic counterparts of the gravitational wave events detected by the Laser Interferometer Gravitational-Wave Observatory (LIGO) [15].

New missions are under preparation, such as Chinese–French SVOM mission [16,17] and eXTP [18,19] or THESEUS mission [20]. SVOM (Space-based multiband astronomical Variable Objects Monitor) mission is planned to be launched at the end of 2021. eXTP (Enhanced X-ray Timing and Polarimetry) mission is currently expected to be launched on 2027. An advanced M class mission concept proposed to ESA within M5 call is THESEUS (Transient High-Energy Sky and Early Universe Surveyor). Its main objective is the study of high-redshift GRBs, as well as the early universe in range of energies from X-ray to gamma-ray, as well as in the near-IR region. The mission was, however, not selected by ESA in the final competition with the other proposed competitive mission but is expected to be reproposed for the next ESA M call (M6) in 2022.

*1.2. X-ray Observer Missions*

X-ray astronomy is especially challenging because one needs to leave the Earth's atmosphere behind to observe X-rays. Many X-ray telescopes were launched to space since 1970s. Currently, nine space telescopes are in operation for astrophysical observation in the X-ray energy range. The oldest working spacecraft is the Chandra and XMM Newton with Wolter I optics [21,22]. Chandra has the finest angular resolution of all currently working high-energy spacecrafts, reaching 0.5 arcsec. The latest launched X-ray telescopes are eROSITA (Extended Roentgen Survey with an Imaging Telescope Array) [23–25] together with ART-XC on the same spacecraft. Summing up all working X-ray observer mission most

common optics is Wolter I type applied onboard Chandra, XMM Newton, SWIFT, Suzaku, NuSTAR, eROSITA and ART-XC instruments (note also the major mission ESA ATHENA in preparation). For Wolter I type optic the nominal FOV is typically 30 arcminutes. Some experiments, however, use simple condenser optics, such as NICER instrument [26], as example of condenser or concentrator optics [21,22]. For high-energy X-rays where the reflective optics is nonoperational, usually coded masks are used, e.g., Integral IBIS. However, the division is not strict, as some observers missions are sometimes able to perform at least limited sky surveying and monitoring.

## 2. Microsatellite Payload Study

The objective of this study is to setup a scientific small satellite mission using a 16U CubeSat for Gamma Ray Burst (GRB) localization and investigation in the X-ray range, with focus on the scientific payload. The study of another types of X-ray transients is also expected.

Since GRBs are relatively bright, these targets are suitable for smaller satellites—there are several mission concepts using nanosatellites. An individual nanosatellite can detect a GRB by using directional detectors. However, for precise GRB source localization, constellations of nanosatellites can be used. The principle of operation of these constellations is to measure the time difference between the arrival of the gamma ray signal at the different satellites. Based on precise timing and known in-orbit positions of the satellites, location of the source in the sky can be determined by triangulation [27]. The satellites will transmit alert data about the detected GRBs, enabling ground-based GRB localization by computation and rapid follow-up observations by other missions, both in space as well as on ground.

The examples of follow-up nanosatellite missions are HERMES, GRID, or CAMELOT missions, with BurstCube or GRBAlpha mission as precursor spacecrafts.

The Italian High Energy Rapid Modular Ensemble of Satellites (HERMES) mission will be a constellation of CubeSats, where each satellite will carry a detector sensitive in the energy range of 2 keV–1 MeV [28]. The demonstration mission is expected to be launched in 2022 and will consist of 7 satellites. The Chinese GRID mission is planned to involve GRB detectors (as secondary payloads) on 10–24 CubeSats [29]. The CAMELOT constellation should include more than nine nanosatellites [30]. The verification of the GRB detector for CAMELOT is planned within the 3U CubeSat VZLUSAT-2 succesfully launched on 13 Jan. 2021. Another CAMELOT detector is placed on a 1U CubeSat GRBAlpha in Q1 of 2021 [31]. Finally, BurstCube is a 6U CubeSat developed by NASA, and will detect GRBs using four CsI scintillators, each with an effective area of 90 cm$^2$ [32].

Prompt follow-up observation is to be performed by pointing the satellite to the observation orientation in less than 30 s. This is because of scientific importance of the observations in the early afterglow stage. We note that the X-ray afterglows mostly last longer, but there is lack of the really prompt observations so we propose to take advantage of small satellite to apply this mode. The technology for such rapid repointing is, however, not generally available. Rapid repointing capability of small and light microsatellites will enable to follow-up the afterglow promptly after the trigger, a feat that cannot be achieved with larger satellites easily. Source localization refinement together with rapid follow-up capabilities also have a potential for a regular detection of electromagnetic counterparts of gravitational wave sources. This ability can therefore provide a critical contribution to our understanding of these exciting phenomena.

To support the described scientific objectives, a 16U CubeSat nanosatellite was selected with respect to the considered X-ray localization refinement telescope/optics (focal length around 350 mm). The selected 16U CubeSat has dimensions of 20 × 20 × 45 cm (see Figure 1). These dimensions are necessary to accommodate both payload telescopes and all the necessary bus components including attitude control, pointing, etc. The proposed CubeSat structure has a highly modular design. CubeSats are a class of small spacecraft of cubic shape and are built to standard dimensions (Units or "U") of 10 × 10 × 10 cm [33,34].

The platform can use commercial elements such as EPS (Electronic Power System), OBC (On-Board Computer), COMM (Communication), customized AOCS (Attitude Orientation and Control System) system, and CubeSat structure. For communication, UHF (uplink and downlink) and S-band (uplink and downlink) are proposed as a baseline. The S-band uplink will provide the ability for software updates needed for tuning the payload software. As an option, an X-band (downlink) will provide a high data rate for downloading the scientific data. A high-maturity AOCS system providing the spacecraft stabilization with attitude pointing precision ±1 arcminute and attitude knowledge of ±30 arcseconds is assumed as a minimum requirement. The final AOCS requirements will depend on the payload FOV and required exposition time. The current astronomical CubeSatellites of similar size, e.g., BRITE (12U) usually achieve attitude determination of 10 arcsec, with attitude control accuracy better than 1.0 deg, with stability of 1 arcmin rms (or FWHM of about 2 arcmin) [35]. Improving the CubeSat stabilization to order of arcsec is challenging but achievable, requiring larger CubeSat (12-16U) and really expensive as precision sensors and actuators are needed [36].

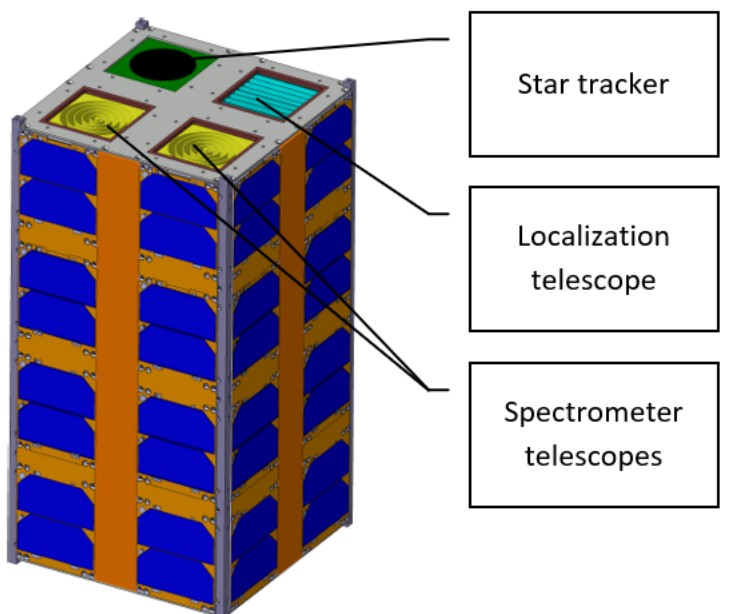

**Figure 1.** The 16U CubeSat with X-ray telescopes payload concept design.

The special feature of having a high-agility spacecraft is needed. It is based on the requirement to start observation as soon as possible from GRB localization reception from third parties, such as GCN Network or LIGO for GWEs or transients detected by monitors described in the introduction chapter. For GRBs, a GCN (The Gamma-ray Coordinates Network, [14]) system is providing fast information on newly detected triggers. To transfer the alert signal to the spacecraft, ground-space real-time communication is needed, such as that provided by Iridium or Globalstar. The mutual spacecraft and Iridium satellite transverse movement cause the Doppler shift, and thus reduce communication coverage. Maximum coverage is attained at high inclinations where the satellite roughly co-orbits with the Iridium network [37]. For CubeSat the communication ability with Iridium network was demonstrated for example by Riot et al. [38] also on low inclinations. The preliminary requirement for CubeSat was set to be able to perform attitude change of 90° by less than 30s in each axis. It is not possible by using a standard AOCS system, which is limited on 1°/s altitude knowledge from star tracker. A special system completing the standard AOSC based on the secondary reaction wheels can allow for this special spacecraft agility feature [36]. This is feasible but expensive due to the high-performance actuators, with each one in the range of 1 million €, and one costs at least 3 million [36].

GRBs are detected randomly, but are relatively frequent. Based on the Fermi GBM Burst Catalog [39], hundreds of GRBs with a flux of about 5 ph cm$^{-2}$ s$^{-1}$ in the energy band 8 keV–40 MeV are detected each year (see Figure 2).

Because of the time needed to accurately determine the GRB position, most afterglow measurements were made hours after the burst, and little is known about the characteristics of afterglows in the minutes following a burst, when the afterglow emission is actively responding to inhomogeneities in both the fireball and the circumburst environment.

The typical GRB X-ray afterglow X-ray fluxes 11 hrs after the GRB trigger are (1–200) $\times$ 10$^{-13}$ erg cm$^{-2}$s$^{-1}$ in the energy band 0.3–10 keV [40]. However, during the period just after GRB (within 100 s), the flux can reach 2 $\times$ 10$^{-9}$ erg cm$^{-2}$s$^{-1}$ as a consequence of an initial steep decay, consistent with it being the tail of the prompt emission from photons that are radiated at large angles relative to our line of sight [41]. However, the scenario is rather complex, as there are cases where bright X-ray flares were observed minutes after the GRB detection [42]. We note that the decay phase of individual bright X-ray flashes may be emissions from the high latitude of a relativistic shell [43,44] and that detections of a large sample of X-ray flares could help to test this hypothesis. Once we require this observational ability for the microsatellite, we state 10 cm$^2$ as a minimum reasonable effective area to achieve the necessary sensitivity limits. This number is set to perform trade-off for suitable optics technologies.

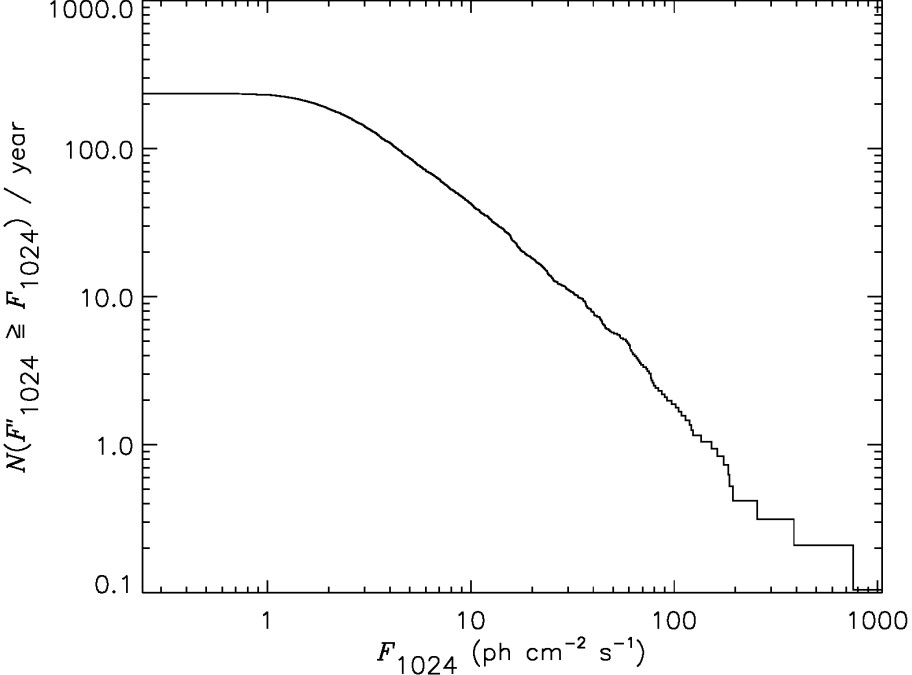

**Figure 2.** Complementary cumulative distribution function of 1024-msec peak fluxes of all Gamma-Ray Bursts (GRBs) detected per year by Fermi/Gamma-ray Burst Monitor (GBM). Note different energy range 8–40 MeV Reprinted with permission from ref. [30] 2022 Society of Photo-Optical Instrumentation Engineers (SPIE).

The proposed 16U CubeSat payload instrumentation represents extension of VZLUSAT-1 and VZLUSAT-2 efforts, with emphasis on high-energy instrumentation and high-energy astrophysics. Mainly, the goal of the proposed satellite is already astrophysical science, not technology testing, as was the case for VZLUSAT-1 and VZLUSAT-2.

## 2.1. X-ray Localization Refinement Telescope/Monitor

The X-ray localization refinement telescope proposed in this study is a telescope based on multifoil Lobster Eye (LE) optics, combined with a low-power X-ray pixel detector (see Table 2). X-ray optical system [45,46] is paired with a semiconductor pixel detector

Timepix [47,48] as focal plane imager (Figure 3). This telescope uses two-dimensional LE optics based on the Lobster Eye principle [49,50]. The telescope provides a relatively wide field of view of 5° × 5°, and a reasonable range of energies (approximately 3 to 7 keV). Timepix is a single-photon-counting detector based on a highly integrated ASIC chip. This type of detector—developed at CERN within the MEDIPIX collaboration [51]—was successfully deployed in orbit [52] and in open space on Proba-V mission [53] and also onboard CubeSat VZLUSAT-1 [54]. The detector is sensitive to X-rays in the range of 3–25 keV with high efficiency [55]. This optics/detector combination was already tested on previous missions, e.g., VZLUSAT-1 and REX1, and the mentioned detector has some advantages for application on-board minisatellites, such as low weight and low cost. The sensitivity range overlap between optics and detector is still reasonable, namely, 3–7 keV.

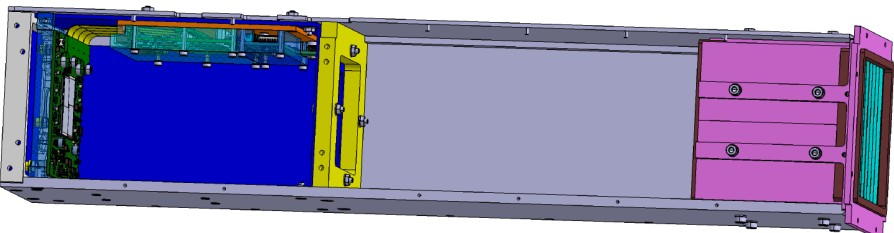

**Figure 3.** X-ray localization refinement/monitor Lobster Eye telescope design concept. Main parts are represented by LE X-ray optics module, including frame (**right**), optical tube, and detector part with electronics board (**left**).

**Table 2.** Parameters of Lobster Eye X-ray localization refinement telescope/monitor system. Given angular resolution is resolution of optics, not taking into account attitude control uncertainty.

| Property | Value |
| --- | --- |
| Telescope outer dimension | $100 \times 100 \times 450$ mm$^3$ |
| Focal length | 355 mm |
| Optical aperture | $69 \times 69$ mm$^2$ |
| Aperture area | 43 cm$^2$ |
| Optics effective area at 4.5 keV | 4.2 cm$^2$ |
| Field of view | $5.8 \times 5.4$ deg$^2$ |
| Angular resolution at 4.5 keV | 4.4 arcmin |

The X-ray localization refinement telescope entrance aperture is based on Lobster Eye (LE) 2D optics module. Its optical aperture is 6.9 × 6.9 cm, which represents 47.6 cm$^2$ of geometrical area. Accounting for the center spacers, this yields an entrance aperture of 43 cm$^2$. The field of view of this system is larger than 5 × 5 degrees. Effective area of such a module is dependent on the photon energy (Figure 4 represents Lobster Eye effective area simulations including direct beams).

The LE optics X-ray image of a point-like astrophysical source is represented by a cross typical for LE optics, as can be seen on Figure 5. The LE telescope picture includes spot as an interpretation of reflected photons in two directions, beams representing only once reflected photons creating the cross around spot, as well as not reflected photons. At low energies of about 1 keV, the photon flux in spot is 2.3 times smaller in comparison to the total flux detected, while a portion of the detected flux is concentrated in the cross. At 4.5 keV, the photon flux in the spot is even 12 times smaller than the total detected photon flux. For this reason, to minimize the observation time to localize the source, picture deconvolution onboard is necessary. This is deemed necessary to make use of photons outside the focal spot (i.e., include all photons, not just the fraction that hits the focal spot).

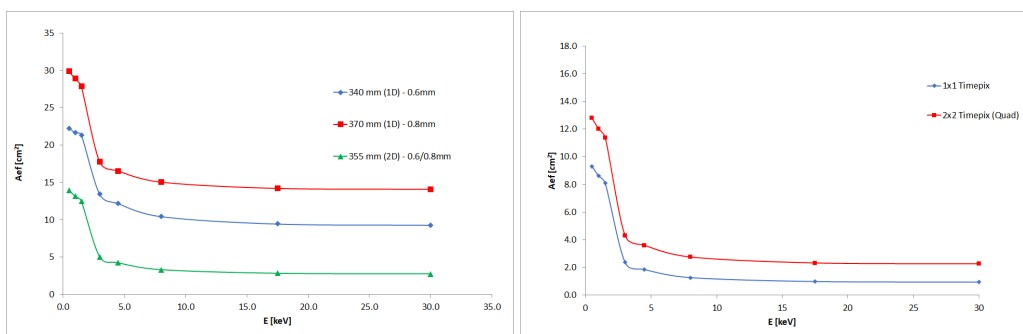

**Figure 4.** Lobster Eye (LE) effective area including direct beams (**left**); effective area of the spot (**right**). Simulations were performed for LE optics module, where first module (1 reflection) of 340 mm focal length and spacers (distances between foils) of 0.6 mm are used. Similarly, results for second 1D module of 340 mm focal length and spacers (distances between foils) of 0.8 mm are used. For full 2D system combining both modules are presented. Analysis uses reflection properties from Figure 6 layer D, 2 nm POLY variant (nonperiodic multilayer on Au basis).

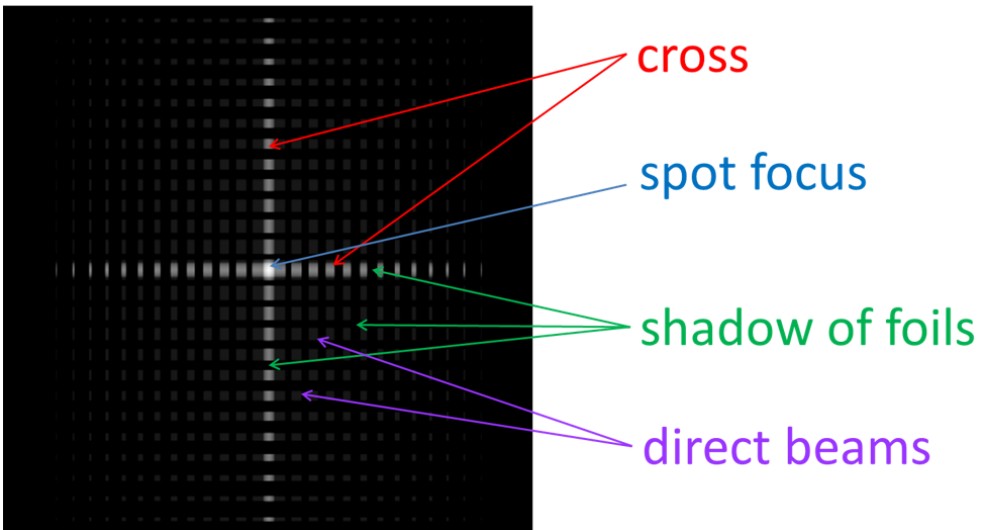

**Figure 5.** LE imaging performance principle. Optics creates main cross with spot focus (twice reflected rays) and also only once, as well as nonreflected direct beam photons. Off-axis source within declared FOV moves image out of center of detector.

The reflection of the foils depends on the material, roughness, and waviness of the reflecting surface. The most commonly used reflective materials for X-ray optics (in general) is gold. The gold reflective layer has, however, a significant decrease around 1 keV (K-edge). For this reason, the new reflective layers/combinations of the reflective layers were studied by Rigaku Prague within the ESA SR-CTP project. A nonperiodic multilayer (based on gold layer) was designed to reach better reflection properties (Figure 6). A suitable nonperiodic multilayer combination was found (layer D—the exact composition of the multilayer Rigaku confidential) that has a 20% better reflectivity than the Au layer (from 2.1 keV to 8.0 keV), with the peak reflectivity being improved by up to 50%. The resulting reflection properties are presented in Figure 6.

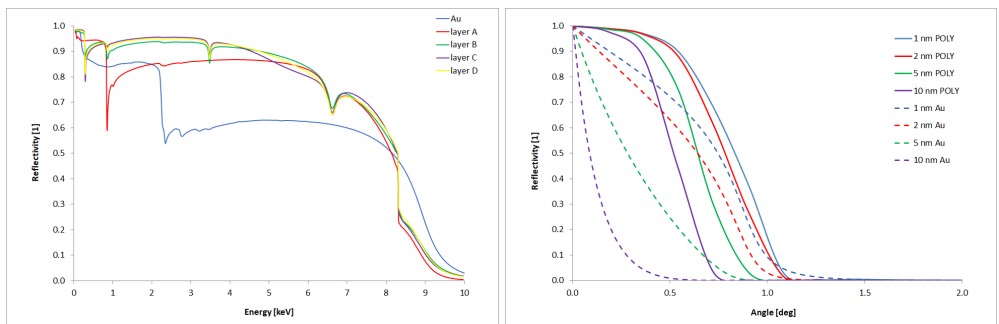

**Figure 6.** Reflectivity dependence on incident photon energy for different multilayers at RMS = 2 nm (**left**). Reflectivity at incident angle of POLY and gold layers for diffrent RMS, 4.5 keV (**right**). POLY—nonperiodic multilayer on Au basis.

As a focal detector, a low-power detector is suitable for cube-satellite application. The Timepix3 noncooled pixel detector (dead time $10^{-7}$ s and power consumption of about 4 W for detector and 8 W for the whole detector system) in quad arrangement was selected as meeting these requirements. This detector operates in the energy range 3–60 keV. The final efficiency of the X-ray telescope shall implement also the detector efficiency as described in [48]. The pixel detector data processing enables to distinguish between photons incident to the detector and background particles forming unwanted signal. The Timepix3 detector is comparable to the CCD or CMOS detectors in standard operation. The new features of the detector include the option to directly stream data reading without necessity of standard reading the full pixel rows. Direct data streaming lower the detector dead-time. This new feature is advantageous for the proposed X-ray mini satellite mission. The detector efficiency in the range 3–7 keV is roughly 90% [56], see also Figure 7.

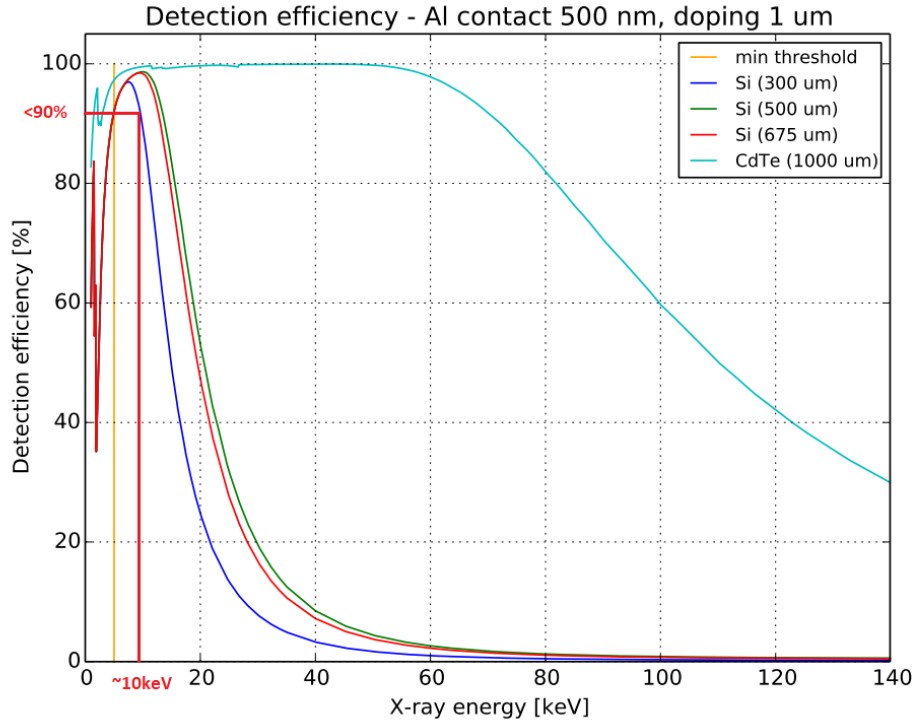

**Figure 7.** Pixel detector efficiency curve Reprinted with permission from [56] 2017 ADVACAM s.r.o.

The proposed observing strategy as well as the astrophysical issues are addressed later in a special section.

## 2.2. The X-ray Spectroscopic Telescope

The X-ray spectrometer information provides complementary information to the light curves or fluxes of GRBs X-ray counterparts, as well as other X-ray astrophysical sources. The finer energy resolution is advantageous for the instrument, e.g., to obtain the red shift of the source and abundances of the elements. The observation requires sufficient flux reaching the cooled X-ray detector. A sufficient flux in the case of GRBs or X-ray transients has values generally not larger than one photon per centimetre square per second. With such faint, low-flux sources, photon concentration is necessary to focus more photons to the detector area. To concentrate the photons, X-ray optics such as Wolter I type can be used. This is a proven technology of two reflection optics used on many missions with about 30″ FOV, where parabolic and hyperbolic mirrors are used. Another option is to use simple one reflection optic for concentrating photons with about 20″ FOV, such as was used on [26]. With using single refection instead of two reflections, the optical system is simpler and less expensive. However, due to the low FOV of this system, it can only be used to implement the refinement telescope described in the previous chapter.

Small satellites such as cube satellites usually have limited focal length and also limited aperture. To fit the 16U CubeSat, a focal length no longer than 400 mm is to be used. We provide the trade-off of five different types of optics (Figure 8): condenser similar to NICER design (Concentrator), widely used Wolter I optics, two-dimensional LE optics, four reflection multifoil optical system (WMFO), and 1-dimensional, two-reflection WMFO (1DWMFO) as a concept. WMFO design is a new theoretical concept design based on planar foils collected in two perpendicular pairs. The concept is a kind of combination of 2D LE principle with segmented submodules formed by two foils of planarized Wolter I. First pair of planar foils of curvatures similar to Wolter I optic creates linear focus and second pair (again with curvatures similar to Wolter I optics) oriented perpendicularly focuses linear focus to spot. The 1DWMFO represents the optics with first pair of WMFO only. This optics therefore focus the photons to the line focus.

The trade-off results are presented in Table 3 and Figure 9. The calculations are based on a nonperiodical multilayer as presented before in Figure 6. These calculated values compare different optical systems with identical focal length of 388 mm.

All these concepts are suitable for the 16U CubeSat satellite and have theoretical maximum effective area exceeding 10 cm$^2$. To be able to compare the various concepts, an identical aperture of about 23 cm$^2$ was used. The selected aperture diameter of 56 mm is the physical limit of total reflection for Concentrator and LE optics at 2 keV meeting the focal length limit. A larger aperture does not increase the effective area of this telescope optics. On the other hand, the Wolter I and WMFO do not reach the total reflection limit of about 50 mm diameter, and a wider aperture can be designed that still adds into the effective area the fixed 388 mm focal length. The graphical interpretation of the trade-off results is in Figure 10. The way how to increase the effective area of considered telescopes is the extension of telescope focal length, however in the Cube satellite case it can be performed by deployable optic system only. The trade-off results for the case where a focal length is 770 mm are presented in the Figure 10.

We conclude from these results that the best option for the spectroscopic telescope is represented by the Wolter I mirror, and that the second best option is represented by the 1DWMFO arrangement.

As a detector, an VITUS H7LE X-ray Silicon Drift Detector with 10 mm$^2$ SDD chip collimated to 7 mm$^2$ area is planned to be used [57]. The spectral detector has area of 7 mm$^2$, which covers all photons reflected from the optics and fits well the X-ray optics working range (0.1–8 keV). The detector efficiency depends on the energy but is reasonably high (65% for 1 keV, 95% for 2 keV and 98% for 3 keV) and suitable for space X-ray instrumentation applications [58,59].

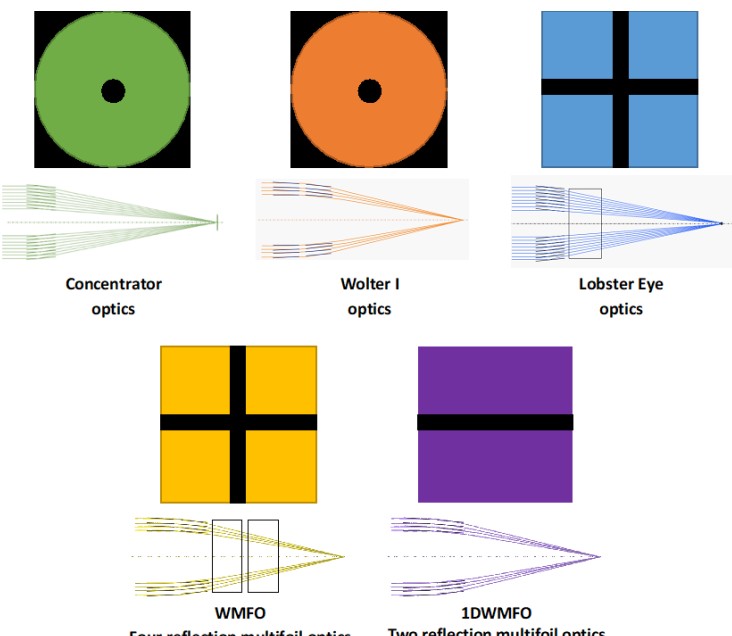

**Figure 8.** Various X-ray spectrometer optical concepts. Black part of aperture cannot be used for focusing. Inner circle or cross is caused by small incident angle requiring extremely long optics close to optical axis. For LE, 8 mm inner region creates ack cross of area reducing aperture area.

**Table 3.** X-ray spectrometer optic concepts trade-off; maximum theoretical effective area versus energy calculation. Aperture area is 23 cm$^2$; focal length 388 mm. Real effective areas will be at least by one third smaller. Reduction is caused by mechanical mounting design and by related shell edges at aperture surface.

|  | Concentrator | Wolter I | LE | WMFO | 1DWMFO |
|---|---|---|---|---|---|
| **System** | 1 reflection | 2 reflections | 2 reflections | 4 reflections | 2 reflections |
| **Incident angle** | alpha/2 | alpha/4 | alpha/2 | alpha/4 | alpha/4 |
| **Foil convergence** |  |  |  |  |  |
| **length [mm]** | 776 | 1552 | 776 | 1552 | 1552 |
| **Focal length [mm]** | 388 | 388 | 388 | 388 | 388 |
| **Diameter [mm]** | 16 to 56 | 16 to 56 | 8 to 56 | 8 to 56 | 12 to 56 |
| **Effective area [cm$^2$]** |  |  |  |  |  |
| **@1keV** | 14.02 | 16.44 | 11.17 | 13.49 | 15.87 |
| **@2keV** | 8.17 | 16.96 | 6.59 | 14.77 | 16.62 |
| **@3keV** | 3.57 | 14.60 | 2.97 | 12.55 | 15.16 |
| **@4keV** | 1.56 | 9.43 | 1.46 | 7.99 | 11.71 |
| **@5keV** | 0.54 | 6.02 | 0.76 | 5.12 | 9.00 |
| **@6keV** | 0.14 | 3.95 | 0.39 | 3.40 | 6.99 |
| **@7keV** | 0.02 | 2.62 | 0.19 | 2.36 | 5.51 |
| **@8keV** | 0.00 | 1.65 | 0.08 | 1.60 | 4.23 |
| **@9keV** | 0.00 | 0.11 | 0.00 | 0.28 | 1.08 |
| **@10keV** | 0.00 | 0.05 | 0.00 | 0.22 | 0.82 |

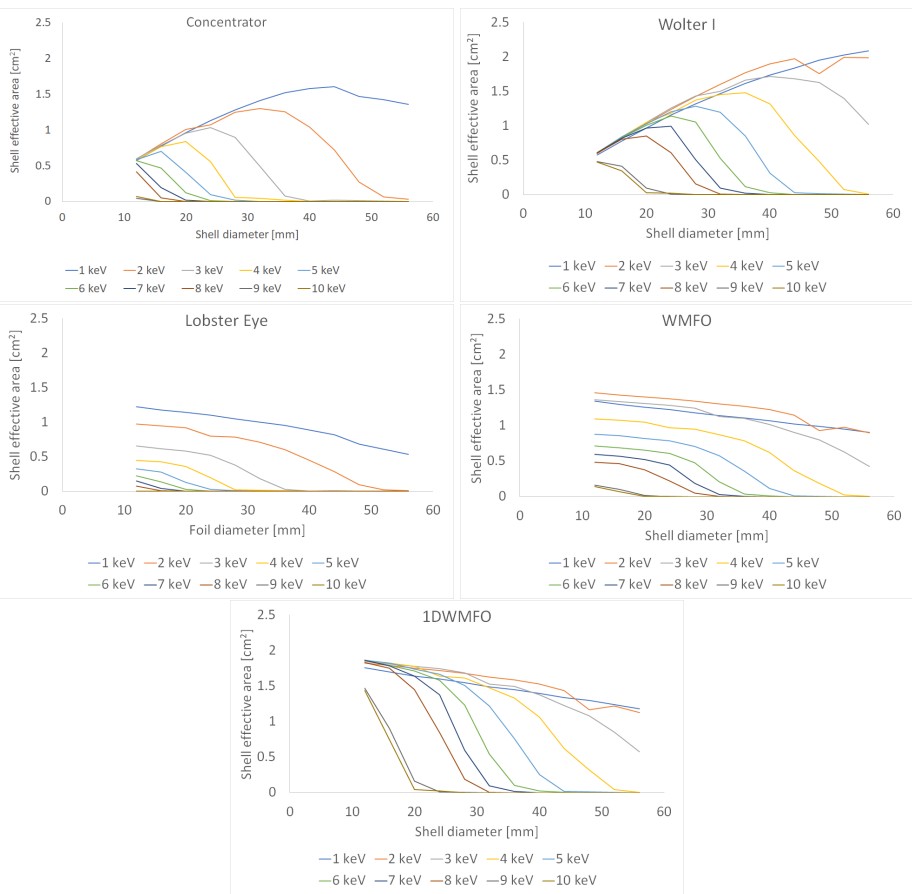

**Figure 9.** X-ray spectrometer optics effective area calculation for particular shell of diameter/dimension respective to optical axis. Final optics properties such as effective area for particular energy are then represented by combination of all shells. In fact, total effective area is represented by simple sum of effective areas of all shells.

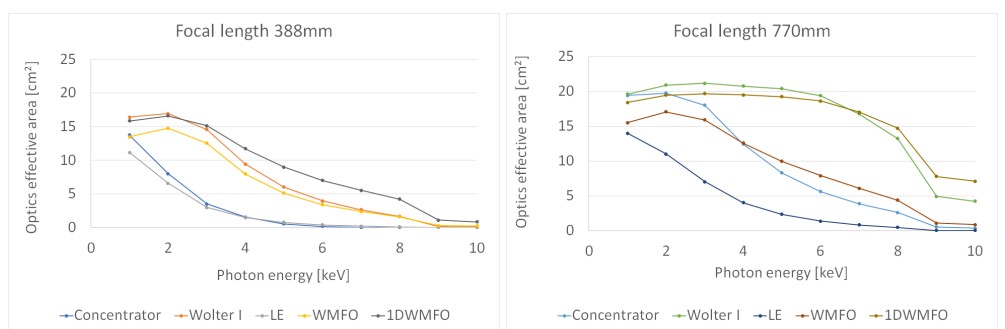

**Figure 10.** X-ray spectrometer optics effective area calculation for particular diameters relative to optical axis.

### 2.3. The Proposed Observational Strategy

Satellites with imaging X-ray telescopes typically perform pointing to selected targets with a limited observing time. This allows observations with high temporal and/or spectral resolution. On the other hand, satellites dedicated to long-term (weeks or months) monitoring of the X-ray sky, including persistent as well as transient sources, are limited. We note that such long-term observations are required for better understanding of physical processes in numerous variable celestial high-energy sources. Relatively inexpensive nanosatellite missions could represent an alternative to long-term monitoring of either full sky (all sky monitors), or alternatively, particular sky areas, e.g., area of the Galactic center or Large Magellanic Cloud with high density of interesting high-energy sources [60].

In addition to the sky monitoring, the satellite has potential to detect X-ray afterglows of gamma-ray bursts.

The proposed LE monitor/telescope is perfectly suited for both responding to newly detected bright transients, as well as for long-term monitoring of preselected area with high density of potential bright and violently variable targets such as the Galactic center. In this mode, the telescope will be continuously pointed at the particular sky area.

For this pointed monitoring mode, mostly X-ray binaries belong to scientifically important targets, both Low-Mass X-ray Binaries (LMXBs) as well as High-Mass X-ray Binaries (HMXBs). These targets can belong both to X-ray transients as well as persistent X-ray sources. For pointed observations and exposure time of 1 ks, the estimated sensitivity of this type of LE telescope/monitor is of order of $5 \times 10^{-10}$ erg/cm$^2$/s [60] consistent with the proposed observational strategy and astrophysical goals.

HMXBs include compact object (neutron star or black hole) and star of an early spectral type (O, B). In HMXBs with the eccentric orbit, the periastron passages may be responsible for X-ray brightenings.

The distribution of X-ray binaries in the galaxy is not homogeneous and this fact can allow optimized monitoring and observing strategy for lobster-eye monitors. Important fact is the concentration of both transient and persistent LMXBs toward the galactic plane and the galactic bulge [61]. On the contrary, HMXBs strongly concentrate toward the galactic plane but not toward the bulge [62]. These distributions are illustrated in Figure 11.

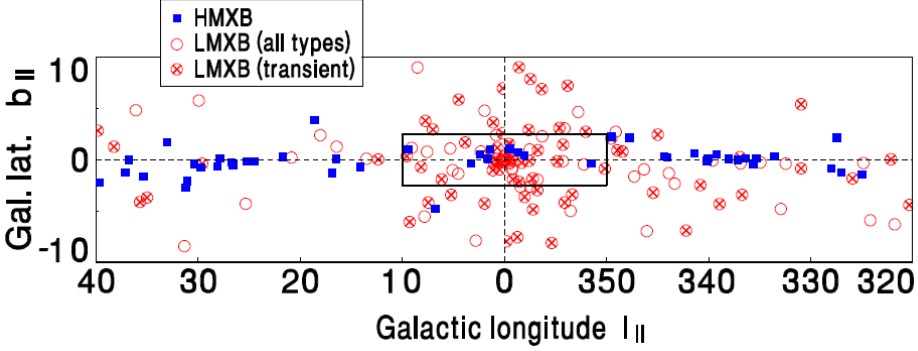

**Figure 11.** Field of center of galaxy (20 × 80 degrees). Positions of known Low-Mass X-ray Binaries (LMXBs) and High-Mass X-ray Binaries (HMXBs) are marked. Most dense accumulation is in galaxy center where LE telescope should be pointed Reprinted from [60].

The galactic plane and galactic center therefore represent valuable areas for long-term monitoring in X-rays. In this type of pointed monitoring mode, there is no need to cover the whole sky. The FOV of the considered monitor will be approximately 5 × 5 degrees, large enough to include positions of numerous potentially interesting targets, if pointed toward the galaxy bulge.

The LE X-ray telescope/monitor directed towards the galaxy center can hence provide the following scientifically valuable observations: (i) A long-term measurement of the light curves of bright persistent X-ray binaries in the FOV, and (ii) detection and measurement of the light curves of bright transient events of X-ray binaries.

The LE X-ray monitor with Timepix detector will be also able to investigate accompanying X-ray spectral variations (changes of the hardness ratios, e.g., when various states of activity are compared). We note the importance of the considered energy range. The proposed optics/detector combination covering the spectral range approximately 3–7 keV is very suitable for observing these sources.

The considered mission represents both a follow-up as well as monitoring project with detection capabilities. The proposed observational strategy is as follows:

- Continuous monitoring of the center (bulge) of our galaxy, i.e., region where there is a high density of highly variable and transient bright X-ray sources such as LMXB,

HMXB, and repeating FRBs (especially those very active [63,64]). In this mode, the new flaring, transient and violently variable events will be recorded, and positions with accuracy of few arcmin will be provided to the onboard spectroscopy telescope;

- Follow-up mode for bright transient triggers provided either by the Lobster telescope or by third party, e.g., GCN system (GRBs) or LIGO (gravitational waves). In addition to that, including the alerts from the planned cube-satellite GRB missions/constellations such as Hermes or CAMELOT described in the previous sections may be novel and beneficial. Note that the positions provided by these CubeSat missions are not expected to exceed one degree, so the position refinement as described will be of a great value. In this mode, the telescope will point to the new flaring and transient events, and in the case of a positive detection (the estimated sensitivities for different exposure times are $10^{-10}$ erg cm$^{-2}$ s$^{-1}$ for 100 s, $10^{-9}$ erg cm$^{-2}$ s$^{-1}$ for 10 s, and $10^{-8}$ erg cm$^{-2}$ s$^{-1}$ for 1 s), positions with accuracy of few arcmin will be provided to the onboard spectroscopy telescope. In both bases, the spectroscopic telescope provides immediately the X-ray spectrum of the trigger with a potential of following the spectral evolution of the particular target.

## 3. Astrophysical Goals and Issues

In addition to the galactic bulge X-ray sources described in previous section, the main astrophysical goals will include the transient and flaring targets described below. A large majority of all GRBs exhibits an X-ray emission. There is also a dedicated separate group of GRB, the XRFs (X-ray flashes), with emission dominating in X-rays. In addition to that, there is a third group of GRB related objects (yet hypothetical), namely, the off-axis observed GRBs (orphan afterglows). All these facts justify the consideration of an independent experiment for monitoring, detection, and analyses of GRBs and others fast X-ray transients in X-rays, as described in this paper. The wide field and fine sensitivity of LE X-ray Monitor described in this contribution make such instrument important tools in study of GRBs and related objects [65].

We note that the small LE based sky X-ray monitors (similar to the LE telescope described here, but in modular concept to achieve larger FOV) also have a potential of independent sky survey in scanning (not pointed) mode [65,66]. Limiting fluxes of $10^{-12}$ erg cm$^{-2}$ s$^{-1}$ can be then achieved for daily scanning observation. The estimated sensitivities for short prompt emission and early afterglows of GRBs is $10^{-11}$ erg cm$^{-2}$ s$^{-1}$ for 1000 s, $10^{-10}$ erg cm$^{-2}$ s$^{-1}$ for 100 s, and $10^{-9}$ erg cm$^{-2}$ s$^{-1}$ for 10 s. The X-ray sky coverage with large FOVs (e.g., FOV of $6 \times 180$ degrees can be potentially assembled on alternative larger spacecraft; note that also fleet of CubeSats with identical LE monitors onboard can increase the FOV significantly) is expected to contribute to various fields of modern astrophysics [66,67] as follows: (1) GRBs: detection rates of nearly 20 GRBs per year can be expected for the prompt X-ray emission of GRBs, assuming the expected GRB rate 300/year; (2) X-ray flashes: detection rates of nearly eight X-ray flashes per year are expected, assuming XRF rate of 100/year; (3) X-ray binaries: vast majority of galactic XRB are expected to be within the detection limits; (4) stars: because of the low X-ray luminosity of ordinary stars, only nearby stars will be observable. We estimate the lower limit of these stars observable by the LE telescope as 600. However, the sampling rate will be sufficient enough to observe sudden X-ray flux increases; (5) supernovae: the LE telescope should be able to detect the theoretically predicted thermal flash lasting for 1000s for the first time. Together with the optical SNe detection rate and estimates of the LE FOV, we estimate the total number of SNe thermal flashes observed by the LE experiment to be 10/year; (6) AGNs: Active Galactic Nuclei will be one of the key targets. LE will be able to monitor the behavior of the large (1000) sample of AGNs; (7) X-ray transients: X-ray transients of various nature are expected to be covered in the case of whole sky coverage for a long time with a limiting flux of about $10^{-12}$ erg cm$^{-2}$ s$^{-1}$. (8) cataclysmic variables (CVs): important classes of CVs for LE telescope monitoring are nonmagnetic Dwarf Novae (DNe), Supersoft X-ray Sources (SSXSs), Classical Novae (CNe), and Polars with soft X-ray

excess; (9) electromagnetic counterparts of Gravitational Wave Events (GWEs); and (10) Fast Radio Bursts (FRB—note the recently found connection with the Galactic Soft Gamma-Ray Repeater (SGR) 1935+2154, and also with a hard X-ray burst [68,69] and references therein). Theoretically, FRBs may be produced by strange quark stars, and X-ray bursts are possible in the process [63].

*GRBs Science and Issues*

While the scientific payload has potential for investigation of variable and transient X-ray sources in general, the GRB science is expected to play an important role.

The position refinement LE X-ray telescope has the potential to observe and to improve the localizations of GRBs coming mostly from third parties, as discussed in previous sections. After that, the X-ray spectroscopic telescope will be promptly (in less than 30 s) pointed to the refined position and perform X-ray spectroscopy.

The X-ray spectroscopy as proposed in this paper may bring interesting knowledge for X-ray afterglows of GRBs. Absorption and reprocessing of GRB radiation in the environment of cosmological GRBs can represent a probe of their progenitors. Transient X-ray emission line and absorption features in the prompt and early afterglows of GRBs are sensitive to the location and density structure of the reprocessing and/or absorbing material. There were only few detections of such features reported in the past, and the significance is mostly marginal. On the other hand, transient X-ray emission lines in these objects were found by recent X-ray satellites, justifying a more detailed theoretical investigation of their origin. The spectral data may hence contribute to general physics constraints on isotropy, homogeneity, and location of the reprocessing material with respect to the relevant GRB sources [70].

## 4. Conclusions

This paper presents a conceptual study of a novel scientific astrophysical payload for a 16U CubeSat microsatellite spacecraft, suitable for study of prompt Gamma-Ray Bursts (GRBs) X-ray afterglow observation with potential to study other types of X-ray transients and X-ray variable sources, both galactic as well as extragalactic. The proposed satellite is equipped with two types of X-ray telescopes, one for localization refinement and monitoring of the astrophysical sources, and the other for the detailed spectroscopic observation of the GRB X-ray afterglows and other transients. The GRB localization refinement telescope uses Lobster Eye (LE) optics with Field-of-View (FOV) of 5° × 5° and Timepix3 quad X-ray pixel detector. For GRB afterglow observation spectroscopic telescope, a trade-off study is provided. All studied designs provide maximum reachable effective area exceeding 10 cm$^2$ at 1 keV. As a new concept the planar Wolter I with two reflections, 1-D Multi-Foil Optics (1DWMFO), and four reflection 2D Multi-Foil Optics (WMFO) are evaluated. For focal length fitting the Cube-Sat focal length of 388 mm the 1DWMFO, Wolter I and WMFO optics reaches maximum theoretical effective area over 15 cm$^2$. For doubled length of 770 mm, implementing a deployable system for the payload, the 1DWMFO, Wolter I and concentrator design reach similar maximum theoretical effective area of around 20 cm$^2$ at 2 keV. The limitation for the optic in this case is defined by the available aperture for 16U CubeSat. We note that the real effective areas will be smaller. This reduction is caused by mechanical design of baffle and shell edges at the aperture area. For the spectroscopic telescope, an X-ray linear detector for 1DWMFO can be used. For other optics, the X-ray SDD detector with energy resolution of about 130 eV can be selected. Based on the provided concept designs, we can conclude that small spacecrafts of CubeSat type represent a promising technique to accommodate short focal length X-ray telescopes for prompt observation of GRBs, as well as other types of astrophysical transients.

We thank the three anonymous referees for their valuable comments and suggestions on how to improve the paper and V. Simon for providing Figure 11.

**Author Contributions:** Individual contributions: Conceptualization, V.D., R.H. and V.M.; methodology, V.D., R.H. and V.M.; mechanical design, K.P.; validation, A.I., V.M.; formal analysis, A.I.; investigation, R.H.; resources, V.M.; data curation, A.I.; writing—original draft preparation, V.D. and R.H.; writing—review and editing, R.H.; visualization, V.M.; supervision, V.D.; project administration, V.M.; funding acquisition, V.D. and L.P. All authors have read and agreed to the published version of the manuscript.

**Funding:** This research was funded by Ministry of Education, Youth and Sports of the Czech Republic, grant number LTAUSA18094, ESA project 40001250020/18/NL/GLC/hh, EU AHEAD2020 project Grant agreement ID 871158, and by the Grant Agency of the Czech Technical University in Prague, grant number SGS21/120/OHK3/2T/13.

**Institutional Review Board Statement:** Not applicable.

**Informed Consent Statement:** Not applicable.

**Data Availability Statement:** Not applicable.

**Conflicts of Interest:** The authors declare no conflict of interest.

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
