# Peer review of "Small Spacecraft Payload Study for X-ray Astrophysics including GRB Science"

_universe, doi:10.3390/universe8030144_

Round 1

Reviewer 1 Report

The paper is certainly an interesting and relevant contribution to a special issue on "Gamma-Ray Bursts: Observational and Theoretical Prospects in the Era of Multi-Messenger Astronomy" but needs extensive rewriting: as it stands it is very confusing and abstruse to read.

  • Section 1: Introduction has no reason to delve into discussing so many previous or ongoing missions, and it does not explicitly mentions detecting GRB, that is to say having triggering capabilities on transients; for instance on table 1 IBIS and AGILE are not the trigger providers for GRB detections, contrary to eg GBM, so table 1 mixes different situation : followers (even if on the same observatory like IBIS/SPI or super-AGILE/AGILE) and detectors like the GBM. In general the triggering capabilities of a U16 nanosat need to be discussed. I ended up confused about whether it is a meant to be a follow up or monitoring project with detection capabilities. The major relevant observatory of the future, ATHENA, is not mentioned. It is with respect to such a major observatory that the addition of lightweight telescopes should be discussed. Finally it seems that the authors have experience on the topic with the VZLUSAT-2 but they do not discuss how the presented 16U nanosat compares to or extend the VZLUSAT-2 effort.
  • section 2 : this is likely the core of the paper content but all the important stuff are missing : values and figures are provided without any clear explanation about how they were obtained; fig.3 shows a CAD design but without any discussion of the subsystems. The simulations are not presented. The exact meaning of the legends for fig.4 and 6 are barely presented in the text, and unless I missed it table 2 is not referenced in the text. At the end of the reading I could not figure out anymore whether an LE or Wolter I optics was to be chosen, and what the tradeoff is. Finally the authors mention here the VZLUSAT-2 saying that on-orbit demonstration will occur at the end of 2021.....
  • section 3: insufficiently clearly present the observation strategy, and delves into observing other types of transients instead of focusing on GRBs, which are fast transients and typically require dedicated efforts. It should just be a subsection of section 2 IMHO.
  • section 4 : has a very long off-topic discussion of non GRB science goals; these should be summarized in the introduction and section 4 should focus on explaining how main figures for GRB science reach are obtained, based on the design and simulations presented on section2.

A final general comment is that many references are missing at places where assertions are made without the reader being able to determine where they come from.

I will be happy to read a new draft, as I am interested in the topic, but as it stands now this paper needs rescope, rewrite, and improvement in details.

Author Response

Section 1: Introduction has no reason to delve into discussing so many previous or ongoing missions, and it does not explicitly mentions detecting GRB, that is to say having triggering capabilities on transients; for instance on table 1 IBIS and AGILE are not the trigger providers for GRB detections, contrary to eg GBM, so table 1 mixes different situation : followers (even if on the same observatory like IBIS/SPI or super-AGILE/AGILE) and detectors like the GBM.

Table 1 explained.

 In general the triggering capabilities of a U16 nanosat need to be discussed.

Added.

 I ended up confused about whether it is a meant to be a follow up or monitoring project with detection capabilities.

Explained.

 The major relevant observatory of the future, ATHENA, is not mentioned.

ATHENA added.

 It is with respect to such a major observatory that the addition of lightweight telescopes should be discussed. Finally it seems that the authors have experience on the topic with the VZLUSAT-2 but they do not discuss how the presented 16U nanosat compares to or extend the VZLUSAT-2 effort.

Added and explained.

section 2 : this is likely the core of the paper content but all the important stuff are missing : values and figures are provided without any clear explanation about how they were obtained; fig.3 shows a CAD design but without any discussion of the subsystems. The simulations are not presented. The exact meaning of the legends for fig.4 and 6 are barely presented in the text, and unless I missed it table 2 is not referenced in the text.

Corrected.

At the end of the reading I could not figure out anymore whether an LE or Wolter I optics was to be chosen, and what the tradeoff is.

Corrected/added.

 Finally the authors mention here the VZLUSAT-2 saying that on-orbit demonstration will occur at the end of 2021.....

Corrected.

section 3: insufficiently clearly present the observation strategy, and delves into observing other types of transients instead of focusing on GRBs, which are fast transients and typically require dedicated efforts. It should just be a subsection of section 2 IMHO.

Changed to be subsection as proposed.

section 4 : has a very long off-topic discussion of non GRB science goals; these should be summarized in the introduction and section 4 should focus on explaining how main figures for GRB science reach are obtained, based on the design and simulations presented on section2.

Added/explained. We also explain that the proposed 16U cubesat mission is mission for X-ray astrophysics in general, including investigation of short events, including GRBs.

A final general comment is that many references are missing at places where assertions are made without the reader being able to determine where they come from.

More references added.

Reviewer 2 Report

In this manuscript, the authors proposed a conceptional payload for a 16U 
CubeSat microsatellite spacecraft, aiming for studying GRB X-ray emissions 
and other types of X-ray transients. The microsatellite is equippted with 
two types of X-ray telescopes, one type for localization refinement and 
monitoring of the astrophysical sources and the other type for detailed 
spectroscopic observations. The study itself is very interesting. The 
microsatellite is of low cost, but can potentially contribute significantly 
to the aforementioned astrophysical fields. Anyway, after carefully reading 
through the whole manuscript, I still found some problems. I would like to 
ask the authors to improve these issues before recommending it for publication. 

Major issues: 

(1) Page 1, in the Abstract, the authors mentioned that the electromagnetic 
    counterparts of Gravitational Wave Events (GWEs) are important goal of 
    their microsatellite. GWEs are also mentioned in the Introduction. However, 
    in their Section 4 (on Page 15), when they detailedly described "Astrophysical goals", 
    GWEs do not appear in the eight items. Since GWEs are really very important 
    phenomena in current astrophysics, and also the microsatellite may hopefully 
    be able to detect them, I believe that the authors need to explicitly include 
    GWEs in the "Astrophyscal goals". 

(2) Many unknown X-ray sources may exist in the Universe. On Page 1, 
    the authors wrote "New astronomical objects of unknown types are also being 
    observed and classified as Exotic X-ray transients. ... many open questions 
    about the details of their physics remain." This is quite true. In fact, 
    fast radio bursts (FRBs) are one type of such unknown sources. First reported 
    by Lorimer D. R. et al. in 2007 (Science, 318, 777), FRBs have attracted the 
    focus of many researchers. Rapid progresses are encouragingly made in recent years. 
    Especially, a fast radio burst was found to be connected with the galactic 
    Soft Gamma-ray Repeater (SGR) 1935+2154 and also associated with a hard X-ray 
    burst (Mereghetti, S., et al. 2020, arXiv:2005.06335; Pearlman, A. B., et al. 2020, 
    arXiv:2005.08410; and references therein). Theoretically, FRBs may be produced 
    by strange quark stars and X-ray bursts are possible in the process (Geng, J., 
    et al. 2021, The Innovation, 2, 100152 (i.e. arXiv:2103.04165) ). Since 
    the FRB study is an extremely active field today, I strongly suggest the 
    authors to add FRBs into the "Astrophysical goals". 

(3) Page 13, Lines 341-346, when discussing goals of pointed monitoring mode,
    in addition to HMXB and LMXB, I think repeating FRBs are also interesting 
    targets (see Geng, J., et al. 2021 above and references therein), 
    especially those very active onces. 

(4) Pages 16--17, in the Reference section, for many reference items, either 
    the volume numbers or the page numbers are missing. For example, in items 
    of Items 18, 35, 41, ... .  

    As an example, the Item 41 is: "Astronomy and Astrophysics, 2021, 647." 
    Seems no page number here. 

    Please carefully check through all the reference items. Also make sure 
    that their formats are consistent with each other and uniform. 

Minor issues: 

(1) Normally, every figure should be mentioned at least once in the main text. 
    However, Figure 1 is not mentioned in the text. 

(2) Page 5, L165, "Attitude Orientation an Control System", "an" --> "and"

(3) Page 6, L197-199, "The typical GRB X-ray afterglow X-ray fluxes 11 hrs 
    after the GRB trigger are (1 - 200) x 10 13cm^-2s^-1 in the energy 
    band 0.3-10 keV. However at the times just after GRB (within 100 sec) 
    the flux can reach 2 x 10^9cm^-2s^-1 ...". Here, the flux unit is cm^-2 s^-1, 
    maybe "erg" is missing in the two flux units ? 

(4) Page 7, Table 2 is not mentioned in the main text. 

(5) Page 7, Figure 4 is not mentioned in the main text. Also, in the last line 
    of the figure caption of Figure 4, "from Figure 6 layer D, 2 nm POLY variant."
    Here "POLY" is not defined. Please define it. Note that "POLY" also appears 
    in Figure 6.  

(6) Page 8, Figure 6, in the right panel, the X-axis is marked as "Angle [degV]". 
    I do not understand why the unit of "angle" is "degV". Are you sure it is not 
    "deg" ? 

    Another problem in this figure: in the caption, the authors wrote "The reflectivity 
    dependence on the incident angle for different multi-layers - RMS = 2 nm (left).".
    However, in the left Panel, the X-axis is "Energy [eV]". So, it is not a dependence 
    on the incident angle, but a dependence on the incident photon energy. 

(7) Page 14, Lines 378-379, "bright X-ray sources such as LMXRB and HMXRB." 
    Note that "LMXRB" and "HMXRB" are not defined. They might should be 
    "LMXB" and "HMXB". 

(8) Page 14, Line 399, "exists which emission dominates in the X-ray spectral range", 
    I am not quite sure, but "which" maybe should be changed to "whose" ?

(9) Page 15, Line 412, "contribute significantly to various fields of modern 
    astrophysics [44][44] as follows." Note that there are two "[44]". 

(10) Page 16, Line 477, "a X-ray linear detector", "a" --> "an". 

Author Response

Major issues:

(1) Page 1, in the Abstract, the authors mentioned that the electromagnetic

    counterparts of Gravitational Wave Events (GWEs) are important goal of

    their microsatellite. GWEs are also mentioned in the Introduction. However,

    in their Section 4 (on Page 15), when they detailedly described "Astrophysical goals",

    GWEs do not appear in the eight items. Since GWEs are really very important

    phenomena in current astrophysics, and also the microsatellite may hopefully

    be able to detect them, I believe that the authors need to explicitly include

    GWEs in the "Astrophyscal goals".

Corrected and added.

(2) Many unknown X-ray sources may exist in the Universe. On Page 1,

    the authors wrote "New astronomical objects of unknown types are also being

    observed and classified as Exotic X-ray transients. ... many open questions

    about the details of their physics remain." This is quite true. In fact,

    fast radio bursts (FRBs) are one type of such unknown sources. First reported

    by Lorimer D. R. et al. in 2007 (Science, 318, 777), FRBs have attracted the

    focus of many researchers. Rapid progresses are encouragingly made in recent years.

    Especially, a fast radio burst was found to be connected with the galactic

    Soft Gamma-ray Repeater (SGR) 1935+2154 and also associated with a hard X-ray

    burst (Mereghetti, S., et al. 2020, arXiv:2005.06335; Pearlman, A. B., et al. 2020,

    arXiv:2005.08410; and references therein). Theoretically, FRBs may be produced

    by strange quark stars and X-ray bursts are possible in the process (Geng, J.,

    et al. 2021, The Innovation, 2, 100152 (i.e. arXiv:2103.04165) ). Since

    the FRB study is an extremely active field today, I strongly suggest the

    authors to add FRBs into the "Astrophysical goals".

FRB added as suggested with relevant references.

(3) Page 13, Lines 341-346, when discussing goals of pointed monitoring mode,

    in addition to HMXB and LMXB, I think repeating FRBs are also interesting

    targets (see Geng, J., et al. 2021 above and references therein),

    especially those very active onces.

Repeating FRBs added as suggested.

(4) Pages 16--17, in the Reference section, for many reference items, either

    the volume numbers or the page numbers are missing. For example, in items

    of Items 18, 35, 41, ... . 

    Corrected.

    As an example, the Item 41 is: "Astronomy and Astrophysics, 2021, 647."

    Seems no page number here.

Corrected.

    Please carefully check through all the reference items. Also make sure

    that their formats are consistent with each other and uniform.

Corrected.

Minor issues:

(1) Normally, every figure should be mentioned at least once in the main text.

    However, Figure 1 is not mentioned in the text.

Corrected.

(2) Page 5, L165, "Attitude Orientation an Control System", "an" --> "and"

Corrected.

(3) Page 6, L197-199, "The typical GRB X-ray afterglow X-ray fluxes 11 hrs

    after the GRB trigger are (1 - 200) x 10 13cm^-2s^-1 in the energy

    band 0.3-10 keV. However at the times just after GRB (within 100 sec)

    the flux can reach 2 x 10^9cm^-2s^-1 ...". Here, the flux unit is cm^-2 s^-1,

    maybe "erg" is missing in the two flux units ?

Corrected.

(4) Page 7, Table 2 is not mentioned in the main text.

Corrected.

(5) Page 7, Figure 4 is not mentioned in the main text. Also, in the last line

    of the figure caption of Figure 4, "from Figure 6 layer D, 2 nm POLY variant."

    Here "POLY" is not defined. Please define it. Note that "POLY" also appears

    in Figure 6. 

Corrected and explained.

(6) Page 8, Figure 6, in the right panel, the X-axis is marked as "Angle [degV]".

    I do not understand why the unit of "angle" is "degV". Are you sure it is not

    "deg" ?

Corrected.

    Another problem in this figure: in the caption, the authors wrote "The reflectivity

    dependence on the incident angle for different multi-layers - RMS = 2 nm (left).".

    However, in the left Panel, the X-axis is "Energy [eV]". So, it is not a dependence

    on the incident angle, but a dependence on the incident photon energy.

Corrected.

(7) Page 14, Lines 378-379, "bright X-ray sources such as LMXRB and HMXRB."

    Note that "LMXRB" and "HMXRB" are not defined. They might should be

    "LMXB" and "HMXB".

Corrected.

(8) Page 14, Line 399, "exists which emission dominates in the X-ray spectral range",

    I am not quite sure, but "which" maybe should be changed to "whose" ?

Corrected.

(9) Page 15, Line 412, "contribute significantly to various fields of modern

    astrophysics [44][44] as follows." Note that there are two "[44]".

Corrected.

(10) Page 16, Line 477, "a X-ray linear detector", "a" --> "an".

Corrected.

Reviewer 3 Report

This manuscript reports the study of a scientific astrophysical payload on the CubeSat nanosatellite platform, which aims observations to prompt GRBs X-ray afterglow and other X-ray transients. Rapid localization and follow-up to X-ray transients are currently in high demand, especially in the next operation period of the gravitational-wave detector LIGO/VIRGO. The paper is well organized and written. I think this manuscript is worth publishing after the authors address the points I raise below.

1. The prompt emission and early afterglow of GRB is a major part of this mission’s scientific goal, the lasting time of these events may be typically shorter than 1000 s. To present the capability of data collection of these events to readers, it is better to give a brief description of the sensitivity with different exposure times (e.g., 1 s or 10 s, etc.).

2. In line 48 on page 2, when “initial part of the afterglow” is mentioned, references on the progress of the early afterglow observation (e.g., Vestrand et al. 2014, Science, 343, 38; Troja et al. 2017, Nature, 547, 425) should be included to address the significance of rapid follow-up in the X-ray band.

3. In line 202 on page 6, the decay phase of individual bright X-ray flares may be emissions from the high latitude of a relativistic shell (e.g., Uhm & Zhang, 2016, ApJ, 824, 16; Geng et al. 2018, ApJ, 862, 115). Detections of a large sample of X-ray flares could help to test this hypothesis.

4. Both terms of “X-ray flares” and “X-ray flashes” are used in the article. If the authors mean the same object, please unify it, otherwise, “X-ray flashes” should be clarified.

Author Response

  1. The prompt emission and early afterglow of GRB is a major part of this mission’s scientific goal, the lasting time of these events may be typically shorter than 1000 s. To present the capability of data collection of these events to readers, it is better to give a brief description of the sensitivity with different exposure times (e.g., 1 s or 10 s, etc.).

 description of the sensitivity with different exposure times inserted.

  1. In line 48 on page 2, when “initial part of the afterglow” is mentioned, references on the progress of the early afterglow observation (e.g., Vestrand et al. 2014, Science, 343, 38; Troja et al. 2017, Nature, 547, 425) should be included to address the significance of rapid follow-up in the X-ray band.

 Included.

  1. In line 202 on page 6, the decay phase of individual bright X-ray flares may be emissions from the high latitude of a relativistic shell (e.g., Uhm & Zhang, 2016, ApJ, 824, 16; Geng et al. 2018, ApJ, 862, 115). Detections of a large sample of X-ray flares could help to test this hypothesis.

 Included.

  1. Both terms of “X-ray flares” and “X-ray flashes” are used in the article. If the authors mean the same object, please unify it, otherwise, “X-ray flashes” should be clarified.

We use the terms “X-ray flares” and “X-ray flashes” asccording to the commonly used terminology by the astronomical X-ray community. The term used depends on the physical nature of the relevant target as well as transient characteristics.

Additional improvements

In addition to the comments by referrees, we have addressed the comments expresed by the Editoria office.

The references were completely rewritten and improved.

Fig. 4 was improved.

Round 2

Reviewer 1 Report

I thank the authors for their attention to the comments made. The draft stands ok as it is, for publication.

Author Response

OK, thanks.

Reviewer 2 Report

In this manuscript, the authors proposed a conceptional payload for a 16U 
CubeSat microsatellite spacecraft, aiming for studying GRB X-ray emissions 
and other types of X-ray transients. The study is very interesting. The 
microsatellite is of low cost, but can potentially contribute significantly 
to astrophysics. The problems in the previous version have been well 
improved. I would like to thank the authors for the revisions. I believe
that this version is ready for publication. 

Below, there are some very minor revisions, which could be incorporated by 
the authors when they upload the final version. 

Minor issues: 

(1) Page 5, Figure 1 still is not mentioned in the text. I thought that 
    the "20cm x 45 cm (see Figure 2)" in Line 163 should actually 
    be "20cm x 45 cm (see Figure 1)". Please check.  

(2) Page 6, Line 216, "X-ray flares cpild help to", here "cpild" maybe 
    should be "could" .

(3) Page 7, it seems that Figure 3 is not mentioned in the main text. 
    Please check it.   

(4) Page 13, The first paragraph of Section 2.3, note that there are 
    two "On the other hand" here (Line 342 and Line 344). You may want 
    to simply remove the second "On the other hand" in Line 344.  

(5) Page 13, Lines 359 --- 360, "For pointed observations and estimated 
    exposure time of 1 ks, the estimated sensitivity of...". Note that 
    there are two "estimated" in the same sentense. The first "estimated" 
    could be removed. 

    Page 13, Line 366, "this fact can allows optimized monitoring"
    allows --> allow 

(6) Page 14, Line 389, "HMXB and repeating FRBs (especially those 
    very active)" . Here "FRBs" could be replaced with "fast radio bursts", 
    since FRBs has not been defined. 

(7) Page 14, Line 409, "all GRBs exhibits a X-ray emission."   a --> an .

    Page 14, Line 420, "The estimated .sensitivities for", should remove 
    the dot "." before "sensitivities". 

    Page 14, Line 422, "and 10^-19ergcm^-2s^-1 for 10 sec". Note that 
    10^-19 here maybe should be 10^-9. Please check. 

(8) Page 15, Line 429, "(3) X-ray binaries. Vst 430 majority of galactic 
    XRB are expected to ...", here, "Vst" maybe should be "Vast".

    Page 15, Line 433, "sufficient enough toobserve sudden" 
    toobserve --> to observe 

    Page 15, Line 462, "There were only few detection's of such features"
    detection's --> detections 

(9) Page 16, Line 495, "We thank ... suggestions how to improve the paper" 
                   --> "We thank ... suggestions on how to improve the paper", 
    i.e., add an "on" here.

Author Response

In this manuscript, the authors proposed a conceptional payload for a 16U

CubeSat microsatellite spacecraft, aiming for studying GRB X-ray emissions

and other types of X-ray transients. The study is very interesting. The

microsatellite is of low cost, but can potentially contribute significantly

to astrophysics. The problems in the previous version have been well

improved. I would like to thank the authors for the revisions. I believe

that this version is ready for publication.

OK, thanks

Below, there are some very minor revisions, which could be incorporated by

the authors when they upload the final version.

Minor issues:

(1) Page 5, Figure 1 still is not mentioned in the text. I thought that

    the "20cm x 45 cm (see Figure 2)" in Line 163 should actually

    be "20cm x 45 cm (see Figure 1)". Please check. 

Corrected.

(2) Page 6, Line 216, "X-ray flares cpild help to", here "cpild" maybe

    should be "could" .

Corrected.

(3) Page 7, it seems that Figure 3 is not mentioned in the main text.

    Please check it.  

Corrected.

(4) Page 13, The first paragraph of Section 2.3, note that there are

    two "On the other hand" here (Line 342 and Line 344). You may want

    to simply remove the second "On the other hand" in Line 344.

Corrected.

(5) Page 13, Lines 359 --- 360, "For pointed observations and estimated

    exposure time of 1 ks, the estimated sensitivity of...". Note that

    there are two "estimated" in the same sentense. The first "estimated"

    could be removed.

Corrected.

    Page 13, Line 366, "this fact can allows optimized monitoring"

    allows --> allow

Corrected.

(6) Page 14, Line 389, "HMXB and repeating FRBs (especially those

    very active)" . Here "FRBs" could be replaced with "fast radio bursts",

    since FRBs has not been defined.

Corrected.

(7) Page 14, Line 409, "all GRBs exhibits a X-ray emission."   a --> an .

Corrected.

    Page 14, Line 420, "The estimated .sensitivities for", should remove

    the dot "." before "sensitivities".

Corrected.

    Page 14, Line 422, "and 10^-19ergcm^-2s^-1 for 10 sec". Note that

    10^-19 here maybe should be 10^-9. Please check.

Corrected.

(8) Page 15, Line 429, "(3) X-ray binaries. Vst 430 majority of galactic

    XRB are expected to ...", here, "Vst" maybe should be "Vast".

Corrected.

    Page 15, Line 433, "sufficient enough toobserve sudden"

    toobserve --> to observe

 Corrected.

    Page 15, Line 462, "There were only few detection's of such features"

    detection's --> detections

Corrected.

(9) Page 16, Line 495, "We thank ... suggestions how to improve the paper"

                   --> "We thank ... suggestions on how to improve the paper",

    i.e., add an "on" here.

Corrected.

This manuscript is a resubmission of an earlier submission. The following is a list of the peer review reports and author responses from that submission.

Round 1

Reviewer 1 Report

The manuscript presents a preliminary microsatellite spacecraft study for prompt observation of transient astrophysical object in X-ray spectrum as Gamma Ray Bursts and other sources. I recommend for publication in Universe journal after some editing suggested below.  The text is in some parts difficult to read and a careful english editing is strongly suggested.

Table 1:

  • BAT 1 - 150 keV —> 15-150 keV
  • BAT FoV 1.4 sr —> half-coded, please specify
  • Fermi/GBM FoV 1 sr —> in line 46 the coverage is quoted as 60% of the sky. Please clarify. 
  • the recent GECAM (but also the less recent Konus-Wind and CALET) might be included

MAXI is missing of any reference, you may mention Matsuoka et al. 2009 PASJ

-THESEUS was unfortunately not selected at the M5 call (but it will likely be proposed again at the next ESA call on 2023, for a launch date of 2039). Please, update the text and put references for THESEUS (Amati et al. 2018, AdSpR, 62,191;  Stratta et al. 2018, AdSpR, 62, 662)

-eXTP is expected to be launched on 2027 (see website https://www.isdc.unige.ch/extp/), please update the text

-SVOM in another important detector for GRBs expected to be launched next year

-line 109: nanosatellite —>  microsatellite(?)

-While the use of “@“ (at) and “x” (times) is fine in plots and tables, it should be avoided in written text

line 91: “As an alternative the Integral spacecraft is using coded mask “ do the authors mean INTEGRAL/IBIS? In this case, this was classified in the previous section dedicated to X-ray prospector missions

line 110: I suggest to put here a coincise general description of 16U CubeSate, as for example the one provided in the first sentence of the conclusion (lines 210-215), and recall Figure 1 that is never mentioned in the text

line 114: and X-band —> an X-band (?)

line 125: effective are —> effective area

line 131: you can recall here Figure 2 never mentioned in the text

line 143: 5x5 arcdegree —> 5x5 square degrees 

line 149: “Main out of spout flux is represented by the cross” —> please rephrase

Table 2: Telescope outer dimension: mm —> mm^3

Optical aperture mm—> mm^2

FoV: deg —> deg^2

line 167-181: this part contains several repetitions previously mentioned, I suggest to start directly from line 181 “To fit 16U CubeSat ….”

line 207: is proposed to used —> to be used (?)

line 212: second —> the second 

Author Response

Dear reviewer,

Thank you for your comments. We went through all your comments in detail and implemented the individual changes. We hope that the newly updated manuscript will be more readable and that your comments are fully implemented.

Please see more details in attachment.

Vladimír Dániel

Reviewer 2 Report

See attached pdf.

Author Response

Dear reviewer,

Thank you for your comments. We very appreciate them, mainly those about wrong mm2 vs cm2. We went through all your comments in detail and implemented the individual changes. We hope that the newly updated manuscript will be more readable and that your comments are fully implemented.

Please see more details in attachment.

Vladimír Dániel

Reviewer 3 Report

I think this is an interesting concept being discussed which embraces a lot of the technologies that are being put into new missions, as well as some new ideas to make things work on the scale of a small satellite. 

However, I find that there would be much benefit if there are a lot more references in the text to both the older missions' papers as papers to explain concepts that are often just mentioned by their acronymn. The problem with that is that this makes the paper a bit lost within the larger discussions and more difficult to understand for those not familiar with all concepts.  

I would like to ask the authors to have a go at correcting that before approving for publication

Author Response

Dear reviewer,

Thank you for your comment. We implemented the individual changes to your comments. We hope that the newly updated manuscript will be more readable and that your comments are implemented.

Vladimír Dániel

Reviewer 4 Report

This paper describes the concept study of instruments for localization and detailed observations for gamma-ray bursts dedicated for the future Cubesats mission. The trade-off study of the configuration of localization and observer telescopes is very well explained. It would be really interesting if such a minitualized X-ray observations can be carried out by the Cubesats. Therefore, I would like to recommend publishing this paper. I hope following comments improve the paper much more.

1. Major comments

1-1) In the abstract line 12, you mention that the X-ray observation telescope has no resolution. It would be better to clarify which resolution you are talking about.

1-2) Introduction section line 54, you refer to the THESEUS mission, but as far as I know this mission was not selected in the current ESA M5 mission. So it might be good if you check the latest status of this mission and modify the description if needed.

1-3) In line 109, you start this study from the 16U Cubesat. I could not find the detailed scientific objective which requires this size of Cubesat. Please clarify why this 16U Cubesat is required to achieve your scientific goal.

1-4) From line 109-117, the basic satellite platform is described. Do you have relevant references which describe the characteristics of those systems ?

1-5) In line 120, you mention that the attitude change of 90 deg by less than 30 s is required and it can be achieved by the special system such as the secondary reaction wheels. Do you have any experimental results to show such a system on the Cubesats can realize such quick attitude control ?

1-6) For the X-ray localization refinement telescope, I guess you need to receive the first alert of the GRB detection with a rough localization information. Do you plan to get such information from other satellites such as Fermi-GBM like satellites ? If so, how do you get such information onboard ?

1-7) For the X-ray localization refinement telescope, why do you use the focal plane detector with the energy range of 3-25 keV, while the telescope is effective only up to 7 keV ? What is this sensitivity range gap ?

1-8) I think readers need to know more details about the figure 3. Could you add more explanation such as the differences of the colors ?

1-9) Does the angular resolution listed in table 2 include the attitude control uncertainty (~1') ?

1-10) Line 152, what is the picture deconvolution ? 

1-11) Around line 159, you explain about the focal plane detector. I think it should also be  mentioned the requirement for the GRB observations and how the Timepix3 pixel detector achieves such a requirement. For example, what is the detailed number of the power consumption of this detector, and what about the requirement from the 16U Cubesats system ? or what is the requirement and expected deadtime of this detector ?

1-12) As for the X-ray observer telescope, the detailed study of the optics system is well described. How much effective area do you need ? I think the design goal should be mentioned somewhere. 

1-13) Could you clarify about the FoV of each optics system you studied ? Is it large enough compared with the attitude control uncertainty (~1') ?

2. Minor comments

2-1) I think you should put the half space between the physical values and the unit, e.g., 30s -> 30 s.

2-2) Could you use GRBAlpha, instead of GRBalpha ?

I am sorry for such long comments and if some of them are besides the point.I hope they help to improve your paper.
Best regards,

Author Response

Dear reviewer,

Thank you for your comments. We went through all your comments in detail and implemented the individual changes. We hope that the newly updated manuscript will be more readable and that your comments are implemented.

Please see more details in attachment.

Vladimír Dániel

Round 2

Reviewer 2 Report

The authors have made a few revisions to the paper, but my overall criticism that this does not represent a mission study, as the title and abstract would suggest, has been ignored entirely. Specifically, there is no discussion of the science that would be enabled by the proposed mission. How many GRBs would be detected? Which scientific questions would be addressed by the spectroscopy? The authors did add a figure from Fermi GBM purporting to show fluxes and GRB counts per year. However, this figure is potentially misleading. First, the reference cited in the text [31] does not contain the figure. It appears to be from an older version of the GBM Burst Catalog, but that is difficult to verify given that no reference is given with the figure. There are two reasons why the figure is problematic. First, these are fluxes at the 1.024s time scale, much shorter than the repointing time of the proposed mission. Secondly, GBM operates at a higher energy range, so that results may not be applicable here. Regardless, there is no discussion of the performance requirement beyond "we state 10cm² as minimum effective area."

The work specifically focuses on improvements to the X-ray optics. However, it is difficult to judge the additional science enabled by these improvements, due to a lack of discussion. It appears to me that standard gold coated Wolter I optics might already achieve the requirement.

Finally, I would like to remark that the response by the authors to my first set of comments was not entirely adequate. Without reference to specific changes in the paper, it is nearly impossible to review those changes. For example, the authors write that a definition of scientific goals and objectives but leave it to me to figure out where that would be. This is notwithstanding the fact that as I write above, there still is no such discussion.

Reviewer 3 Report

I have marked up the PDF copy of the text, mainly with suggestions for improvement of the english. 

I think you need to mention Swift (which has a GRB detector) and slews very fast (using a fast slewing technology). I suspect the technology is not accessible. 
